# Automated cell-type classification in intact tissues by single-cell molecular profiling

**Monica Nagendran[1,2†], Daniel P Riordan[3†‡], Pehr B Harbury[3]\*, Tushar J Desai[1,2]\***

[1]Department of Internal Medicine, Division of Pulmonary & Critical Care, Stanford University School of Medicine, Stanford, United States; [2]Institute for Stem Cell Biology and Regenerative Medicine, Stanford University School of Medicine, Stanford, United States; [3]Department of Biochemistry, Stanford University School of Medicine, Stanford, United States

**\*For correspondence:**
harbury@stanford.edu (PBH);
tdesai@stanford.edu (TJD)

[†]These authors contributed equally to this work

**Present address:** [‡]10X Genomics, Inc., San Francisco, United States

**Abstract** A major challenge in biology is identifying distinct cell classes and mapping their interactions in vivo. Tissue-dissociative technologies enable deep single cell molecular profiling but do not provide spatial information. We developed a proximity ligation in situ hybridization technology (PLISH) with exceptional signal strength, specificity, and sensitivity in tissue. Multiplexed data sets can be acquired using barcoded probes and rapid label-image-erase cycles, with automated calculation of single cell profiles, enabling clustering and anatomical re-mapping of cells. We apply PLISH to expression profile ~2900 cells in intact mouse lung, which identifies and localizes known cell types, including rare ones. Unsupervised classification of the cells indicates differential expression of 'housekeeping' genes between cell types, and re-mapping of two sub-classes of Club cells highlights their segregated spatial domains in terminal airways. By enabling single cell profiling of various RNA species in situ, PLISH can impact many areas of basic and medical research.
DOI: https://doi.org/10.7554/eLife.30510.001

## Introduction

In parallel with the development of single-cell RNA sequencing (scRNA-seq), there have been rapid advances in single-molecule in situ hybridization (smISH) techniques that localize RNAs of interest directly in fixed cells (*Shah et al., 2016*; *Huss et al., 2015*; *Chen et al., 2015*; *Wang et al., 2012*; *Larsson et al., 2010*; *Raj et al., 2008*; *Femino et al., 1998*). These smISH techniques involve hybridization of fluorescently-labeled oligonucleotide probes, typically 24–96 per gene, to mark individual RNA molecules with a discrete, diffraction-limited punctum that can be quantitatively analyzed by fluorescence microscopy. smISH has been used in cultured cells to study the subcellular distribution of RNAs (reviewed in [*Buxbaum et al., 2015*]), the consequences of stochastic noise on gene expression (*Raj et al., 2010*; *Raj et al., 2006*), and the impact of cell shape and environment on expression programs (*Moffitt et al., 2016a*; *Battich et al., 2015*).

An increasingly important application for smISH is the simultaneous localization of customized panels of transcripts in tissue, which is used to validate putative cell subtypes identified by scRNA-seq studies (*Grün and van Oudenaarden, 2015*). Performing smISH in intact tissue can also reveal the spatial relationship between the cells expressing secreted signaling factors and the cells expressing the corresponding receptors, information that current scRNA-seq approaches cannot resolve because they require tissue dissociation with irretrievable loss of spatial context. Finally, when applied on a genome-wide scale in tissues, smISH has the potential to entirely bypass scRNA-seq as an upfront discovery tool.

**eLife digest** The human body contains several hundred types of specialized cells that have different roles. The cells form tissues, and each tissue can only work if it has the right cells and if they are correctly organized and distributed to build working structures. This is how the same body parts, e.g. lungs, brains, hearts, end up looking and working the same way in almost everyone.

Cells organize themselves into tissues by exchanging short-range messages between nearby cells. Understanding how cells communicate to form, maintain and repair tissues is a challenge for biologists. Finding ways to examine different signals at the same time would improve our understanding of these important processes.

Now, Nagendran, Riordan et al. developed a microscopy technique that can tackle this issue. The cells can be stained and tagged with already established dyes and markers to measure the location and signals of all cell groups at the same time. By calculating how these cells are distributed in space it is then possible to estimate how they interact with each other. Based on this, Nagendran, Riordan et al. then successfully tested their tool in lung tissues from mice and humans.

This cheap, high-speed technology works on tissue samples from any animal, including humans, and can be easily combined with existing technologies and so be adapted for a wide range of uses. A deeper knowledge of how combinations of signals guide tissue formation and maintenance could help us to better understand what causes developmental diseases, organ failures and cancer. Tools like this could even help to identify key targets for new treatments.

DOI: https://doi.org/10.7554/eLife.30510.002

The development of multiplexed smISH for use in tissue has been challenging due to autofluorescent background and light scattering (*Shah et al., 2016*; *Sylwestrak et al., 2016*; *Moffitt et al., 2016b*; *Chen et al., 2016*; *Choi et al., 2014*; *Lyubimova et al., 2013*). One strategy for addressing this problem is to amplify probe signals by the hybridization chain reaction (HCR, reviewed in (*Choi et al., 2016*); see also (*Wang et al., 2012*) for branched-DNA amplification), which provides up to five orthogonal detection channels. Higher levels of multiplexing can be achieved by repeated cycles of RNA in situ hybridization followed by a re-amplification step (*Shah et al., 2016*), but because a single round of probe hybridization in tissue sections takes hours, multiplexing with HCR is laborious. Unamplified smISH techniques have the practical advantage that hundreds of endogenous RNA species can be barcoded in a single reaction, and then read out with rapid label-image-erase cycles (*Moffitt et al., 2016b*; *Moffitt et al., 2016c*), but these do not provide adequate signal in tissues.

Ideally, a technique for high-throughput profiling would combine all of the RNA probe hybridization and signal amplification steps into a single reaction. Previously, Nilsson and colleagues presented an elegant enzymatic solution to this problem (*Larsson et al., 2010*; *Ke et al., 2013*). They used barcoded padlock probes to label cDNA molecules in cells and tissues, and rolling-circle amplification (RCA) to transform the circularized probes into long tandem repeats. The approach worked in tissues and handled an unbounded number of orthogonal amplification channels. The only limitations were that the RNA-detection efficiency was capped at about 15% (each transcript could only be probed at a single site because the 3' end of the cDNA served as the replication primer), and that the approach required an in situ reverse transcription step with specialized and costly locked nucleic-acid primers.

Here, we report an in situ hybridization technique with performance characteristics that enable rapid and scalable single-cell expression profiling in tissue. Our approach is a simplified variant of the padlock/RCA technique which replaces padlock probes with RNA-templated proximity ligation (*Söderberg et al., 2006*; *Frei et al., 2016*) at Holliday junctions (*Labib et al., 2013*); hence, we term it proximity ligation in situ hybridization (PLISH). As demonstrated below, PLISH generates data of exceptionally high signal-to-noise. Multiplexed hybridization and signal amplification of all target RNA species is carried out in a single parallel reaction, and the RNAs are then localized with rapid label-image-erase cycles. PLISH exhibits high detection efficiency because it probes multiple sites in each target RNA, and high specificity because of the proximity ligation mechanism. PLISH utilizes only commodity reagents, so it can be scaled up inexpensively to cover many genes. It works

well on conventional formalin-fixed tissues that have been cryo- or paraffin-embedded, and can be performed concurrently with immunostaining, making it extremely versatile. Using the murine lung as a characterized model tissue, we show that multiplexed PLISH can rediscover and spatially map the distinct cell types of a tissue in an automated and unsupervised fashion. An unexpected discovery from this experiment is that murine Club cells separate into two populations that differ molecularly and segregate anatomically. PLISH constitutes a novel, single cell spatial-profiling technology that combines high performance, versatility and low cost. Because of its technical simplicity, it will be accessible to a broad scientific community.

## Results

### Proximity ligation in situ hybridization (PLISH)

Proximity ligation at Holliday junctions offers a simple mechanism for the amplified detection of RNA (*Labib et al., 2013*). First, a transcript is targeted with a pair of oligonucleotide 'H' probes designed to hybridize at adjacent positions along its sequence (*Figure 1A*). The left H probe includes a single-stranded 5' overhang while the right probe includes a 3' overhang. Importantly, target RNAs can be tiled with H probe pairs at multiple sites, which is critical for efficient detection of low abundance transcripts (*Figure 1B*). The overhangs are then hybridized to 'bridge' and linear 'circle' oligonucleotides with embedded barcode sequences to form a Holliday junction structure, after which ligation at the nick sites creates a closed circle. Finally, the 3' end of the right H probe primes rolling-circle replication, which generates a long single-stranded amplicon of tandem repeats. Addition of fluorescently-labeled 'imager' oligonucleotides complementary to the barcodes generates an extremely bright punctum at the site of each labeled transcript. Because each barcode sequence is unique, the puncta derived from different target RNAs can be labeled with different colors (*Figure 1C*).

To implement PLISH, we adapted protocols for antibody-based proximity ligation (*Söderberg et al., 2006*). The technique utilizes conventional oligonucleotides, two commercially available enzymes, and procedures familiar to molecular biologists. The ligase and polymerase enzymes are less than half the size of an immunoglobulin G, and they diffuse at least as rapidly as the 60mer DNA hairpins used for HCR amplification (*Choi et al., 2014*; *Joubert et al., 2003*; *Lapham et al., 1997*; *Modrich et al., 1973*). Our initial studies produced bright puncta that were absent if any of the oligonucleotide or enzyme reagents was withheld. The signal from the individual RCA amplicons exceeded cellular and tissue fluorescence background by more than 30-fold, rendering autofluorescence inconsequential (*Figure 1—figure supplements 1A–B* and [*Jarvius et al., 2006*; *Blab et al., 2004*]). Histograms of puncta intensities fit to a negative binomial distribution, as expected for a DNA replication process that terminates stochastically and irreversibly (*Figure 1—figure supplement 1C–F*). The coefficients of variation for the puncta intensity distributions were typically between one and two.

### Highly specific and sensitive detection of RNA transcripts

The requirement for coincident hybridization of two probes at adjacent sites in an RNA transcript should make PLISH highly specific. To evaluate this, we performed several experiments. First, we used PLISH to detect the transcription factor SRY-box 4 (SOX4) in cultured HCT116 cells. A pool of ten H probesets exhibited much higher RNA detection efficiency than a single H probeset, as expected (*Figure 1D*). However, when the RNA-recognition sequence of either the left or right H probe in each set was scrambled, there were no detectable puncta. Thus, both H probes had to be correctly targeted to generate a signal. Second, we tested the sequence-specificity of the PLISH signal in tissue by pre-incubating samples with antisense 'blocking' oligonucleotides complementary to the target RNA at the H probe hybridization sites. For these experiments, we stained mouse lung sections for secretoglobin 1a1 (Scgb1a1), a marker of airway Club cells. Antisense oligonucleotides drastically attenuated the number of PLISH puncta, whereas scrambled blocking oligonucleotides of the same length had no apparent effect (*Figure 1E*). Third, we analyzed murine lung sections for the co-localization of the mRNA transcript and protein product of surfactant protein C (Sftpc), which is expressed in alveolar epithelial type II (AT2) cells. Of the cells that were positive for PLISH signal, 98.5% were also positive for antibody staining (n = 184, *Figure 1—figure supplement 2*). This level

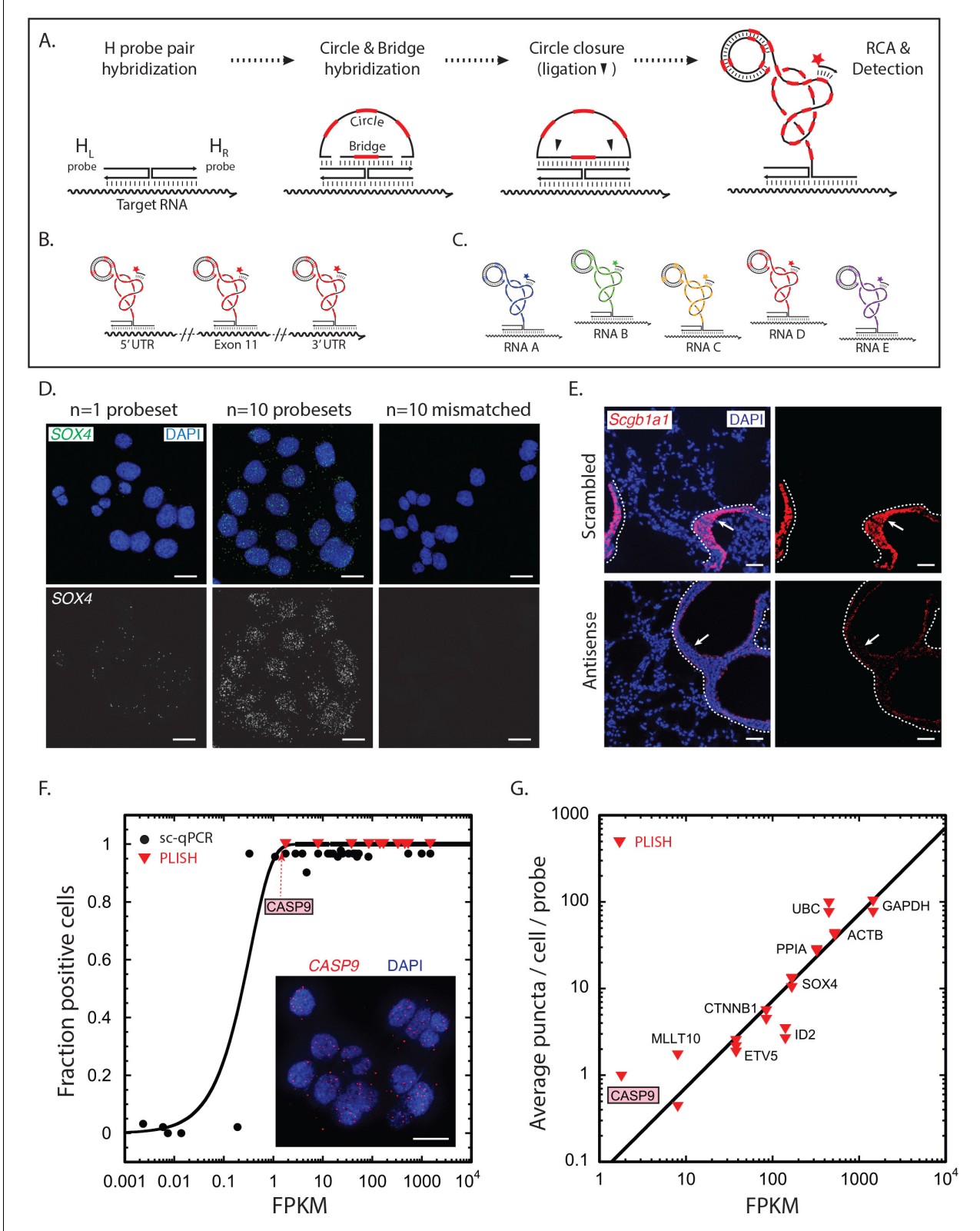

**Figure 1.** RNA detection by proximity ligation at RNA-DNA Holliday junctions. (**A**) Mechanism of RNA detection by PLISH. Left ($H_L$) and right ($H_R$) DNA 'H' probes targeting adjacent sites hybridize to a target RNA. Subsequent addition of circle and bridge oligonucleotides harboring a specific 'barcode' sequence (red dash) results in an RNA-DNA Holliday junction. The nicks in the junction are then sealed by ligation to create a covalently closed circle, and rolling-circle amplification (RCA) generates complementary tandem repeats. The single-stranded amplicons are detected with a fluorescently-

*Figure 1 continued on next page*

*Figure 1 continued*

labeled 'imager' oligonucleotide (red star) that is complementary to the specific 'barcode' sequence. See note in 1C. (**B**) To increase efficiency of detection for low abundance transcripts, H probe pairs embedded with the same barcode sequence can be 'tiled' along the length of the target mRNA. (**C**) Up to five distinct transcripts can be simultaneously detected using five different barcode sequences (one unique sequence for each RNA), and five complementary imager oligonucleotides that are conjugated to spectrally-distinct fluorophores. (**D**) PLISH RNA detection requires coincident hybridization of both left and right H probes, and redundant probesets increase detection efficiency. Cultured HCT116 cells were stained with one (left panel) or ten (middle panel) pairs of H probes targeting the *SOX4* transcript. Many more RNA molecules are detected with ten probesets. No puncta are observed when the RNA-recognition sequences of the left H probes are scrambled (right panel). SOX4, SRY-box 4; Scale bar, 10 μm. (**E**) PLISH RNA detection in tissues is highly sequence-specific. Mouse lung was hybridized with a single pair of H probes targeting nucleotides 228–268 of the *Scgb1a1* transcript. The section in the bottom row was pre-incubated with a 60-base antisense blocking oligonucleotide complementary to nucleotides 219–278, whereas the section in the top row was pre-incubated with a scrambled 60-base blocking oligonucleotide. The antisense blocking oligonucleotide dramatically reduces the *Scgb1a1* signal (bottom), whereas the scrambled blocking oligonucleotide has no effect (top). Note that the PLISH signal is tightly restricted to the bronchial Club cells (arrow). The dashed lines indicate the basal surface of bronchial epithelium. Scgb1a1, secretoglobin family 1A1 member 1; Scale bar, 40 μm. (**F**) PLISH RNA detection sensitivity in cultured cells matches single-cell qPCR sensitivity. FPKM values for 36 mRNAs are plotted against the fraction of HCT116 cells in which they were detected by single-cell qPCR (*Wu et al., 2014*) (filled black circles) or by PLISH (red inverted triangles). The black line is the prediction of a Poisson sampling model for the fraction of cells with at least one transcript, assuming that the transcript abundance increases proportionally with FPKM, and that one FPKM unit corresponds to 2.5 copies per cell. The inset shows PLISH staining for *CASP9*, which has an FPKM value of 2. CASP9, caspase 9; FPKM, Fragments Per Kilobase of transcript per Million mapped reads; qPCR, quantitative reverse transcription polymerase chain reaction; Scale bar, 20 μm. (**G**) RNA abundances measured by PLISH and by single-cell RNA sequencing are highly correlated. A log-log plot showing the average single-cell FPKM value for 10 mRNAs in HCT116 cells (*Wu et al., 2014*) plotted against the number of puncta per cell per probe measured by PLISH. Multiple points at each FPKM value are independent experiments. The data fit to a line of slope 1 with $R^2 = 0.8$.

DOI: https://doi.org/10.7554/eLife.30510.003

The following figure supplements are available for figure 1:

**Figure supplement 1.** Signal-to-noise in PLISH images.
DOI: https://doi.org/10.7554/eLife.30510.004
**Figure supplement 2.** Benchmarking PLISH specificity against a validated antibody by co-staining in tissue.
DOI: https://doi.org/10.7554/eLife.30510.005
**Figure supplement 3.** Estimating efficiency of PLISH probes in tissue.
DOI: https://doi.org/10.7554/eLife.30510.006

of specificity is excellent relative to HCR-amplified smISH, where off-target binding of hybridization probes can account for a quarter of the observed puncta (*Shah et al., 2016*).

To quantify the sensitivity and accuracy of RNA detection, we benchmarked PLISH measurements against a reference-standard dataset of single-cell, quantitative reverse transcription polymerase chain reaction (qPCR) and RNA-seq measurements on HCT116 cells (*Wu et al., 2014*). For genes with fragment-per-kilobase-per-million-read (FPKM) values greater than one, the single-cell qPCR technique detected mRNA in >90% of the cells (*Figure 1F*). However, the fraction of transcript-positive cells dropped quickly between FPKM values of 1 and 0.1. A fit of the qPCR data to a Poisson sampling model suggested that an FPKM value of one corresponded to 2.5 copies per cell (see also [*Marinov et al., 2014*; *Battich et al., 2013*]). The PLISH technique detected RNA transcripts with a sensitivity comparable to single-cell qPCR. For example, Caspase-9 (CASP9) has an FPKM value of 2, and it was observed in 100% of the cells by PLISH. We detected an average of 8 puncta per cell, which is consistent with the prediction of 5 copies per cell from the fit to the qPCR data (*Figure 1F*, inset). For a set of ten genes covering the full spectrum of expression levels in HCT116 cells, the number of PLISH puncta per cell correlated with bulk FPKM values (*Figure 1G*).

To quantify RNA-detection efficiency in tissue, we marked a set of axin 2 (Axin2) transcripts in mouse lung sections using an HCR-amplified smISH procedure (*Choi et al., 2014*) and then determined the fraction of the marked transcripts that could be identified by PLISH. We chose the Axin2 gene because of its low expression level in the lung. HCR detected a sparse population of cells with one to two puncta each (the HCR detection efficiency was low because we used a single HCR probe rather than 24). PLISH puncta generated with a pool of four H probe pairs co-localized with 32% of the HCR puncta (*Figure 1—figure supplement 3*). Thus, the four PLISH probesets detected *Axin2* transcripts with a composite efficiency of 32% and an average per-site efficiency of 9%. This probe efficiency matches or exceeds that of other smISH techniques. The PLISH detection efficiency can be tuned on a per gene basis by altering the number of H probe pairs. Decreasing the number of

probesets pro-rates the number of puncta from highly-expressed genes, while increasing the number of probesets can facilitate sensitive detection of very low-abundance transcripts.

## Visualization of molecular and histological features in tissue

We next characterized the performance of PLISH for low-plex RNA localization in tissues. This experimental format uses a disposable hybridization chamber that is sealed to a coverslip or slide surrounding a tissue section (*Figure 2A*). PLISH detection of up to 5 RNA species is accomplished by stepwise application of reagents through the inlet and outlet ports of the chamber. The puncta from each RNA species are then labeled in a unique color by hybridization to 'imager' oligonucleotides with spectrally-distinct fluorophores. After imaging, the fluorescence micrographs are interpreted by direct visual inspection. We aimed to test whether PLISH provides single-molecule and single-cell resolution in tissues, whether it robustly detects low-abundance RNA species, whether the spatial distribution of RNA is consistent with prior knowledge, whether PLISH is compatible with simultaneous immunostaining, and whether it is compatible with formalin-fixed, paraffin-embedded (FFPE) samples.

First, we analyzed murine lung sections for RNA expression of the ciliated-cell marker Forkhead box J1 (Foxj1), and the Club-cell marker Scgb1a1. *Foxj1* is a low-abundance transcript with an FPKM value of 10 in ciliated cells, as measured by scRNA-seq (*Treutlein et al., 2014*). We observed single cells with multiple discrete *Foxj1* puncta in the terminal bronchiolar epithelium, surrounded by numerous strongly *Scgb1a1* positive cells (*Figure 2B*). These data establish PLISH's single-molecule and single-cell resolution in tissues, and its ability to detect low-abundance transcripts.

Second, we analyzed human lung FFPE sections for RNA expression of *SCGB1A1*, and for protein expression of the basal cell marker, Keratin 5 (KRT5). To do this, we appended two antibody incubation steps to the standard PLISH protocol. Strongly *SCGB1A1* positive cells were localized to the lumen of the airways, overlying KRT5 positive cells (*Figure 2C*), matching the known anatomical distribution of Club and basal cells, respectively. These data establish PLISH's compatibility with simultaneous immunostaining, and with FFPE samples.

Third, we analyzed murine lung sections for RNA expression of three genes: the AT2 cell marker Sftpc, the macrophage-enriched marker Lysozyme 2 (Lyz2), and Scgb1a1. Overlays of the three channels provided a striking visual depiction of the different cell types. Macrophages were bright in the *Lyz2* channel, but absent in the other channels (*Figure 2D*). AT2 cells were bright in the *Sftpc* channel, moderately bright in the *Lyz2* channel and absent in the *Scgb1a1* channel (*Figure 2D*, white cells in the overlay). Club cells were very bright in the *Scgb1a1* channel, but otherwise absent (*Figure 2E*). Finally, putative bronchioalveolar stem cells (*Kim et al., 2005*) (BASCs) were bright in the *Sftpc* channel with a weak punctate signal in the *Scgb1a1* channel (*Figure 2E*). Thus, raw PLISH data can be interpreted without any computational processing, made possible by PLISH's exceptional signal-to-noise in tissues.

We also evaluated how PLISH performs in primary samples of diseased human tissue, to assess whether it will be useful for molecular analysis of the many human diseases that cannot be accurately modeled in animals. One example is idiopathic pulmonary fibrosis (IPF), a fatal lung disease of unknown pathogenesis (*Travis et al., 2013*). The diagnosis of IPF is based on the presence of specific histological features, including clusters of spindle-shaped fibroblasts, stereotyped 'honeycomb' cysts, and epithelial cell hyperplasia. In this regard, single-cell profiling approaches that operate on dissociated tissue (*Xu et al., 2016*) are intrinsically limited because they cannot correlate molecular data with cytologic and spatial features. As a preliminary test, we used PLISH to analyze RNA expression of the AT2 cell marker SFTPC in resected lung tissue from control and IPF patients. In contrast to the uniformly cuboidal *SFTPC*-expressing AT2 cells distributed throughout alveoli of non-IPF lungs (*Figure 2F*), we observed clusters of $SFTPC^{Hi}$ cells of heterogeneous size and varying degrees of flattening lining the airspace lumen of IPF lungs. Surprisingly, many cells that did not appear to be epithelial (i.e., they were not lining an airway lumen) expressed *SFTPC* at low levels. Based on this pilot experiment, PLISH should be a suitable tool for building atlases of RNA expression in human disease. The PLISH data can be overlaid with monoclonal antibody staining patterns that are the mainstay of pathologic diagnosis and classification.

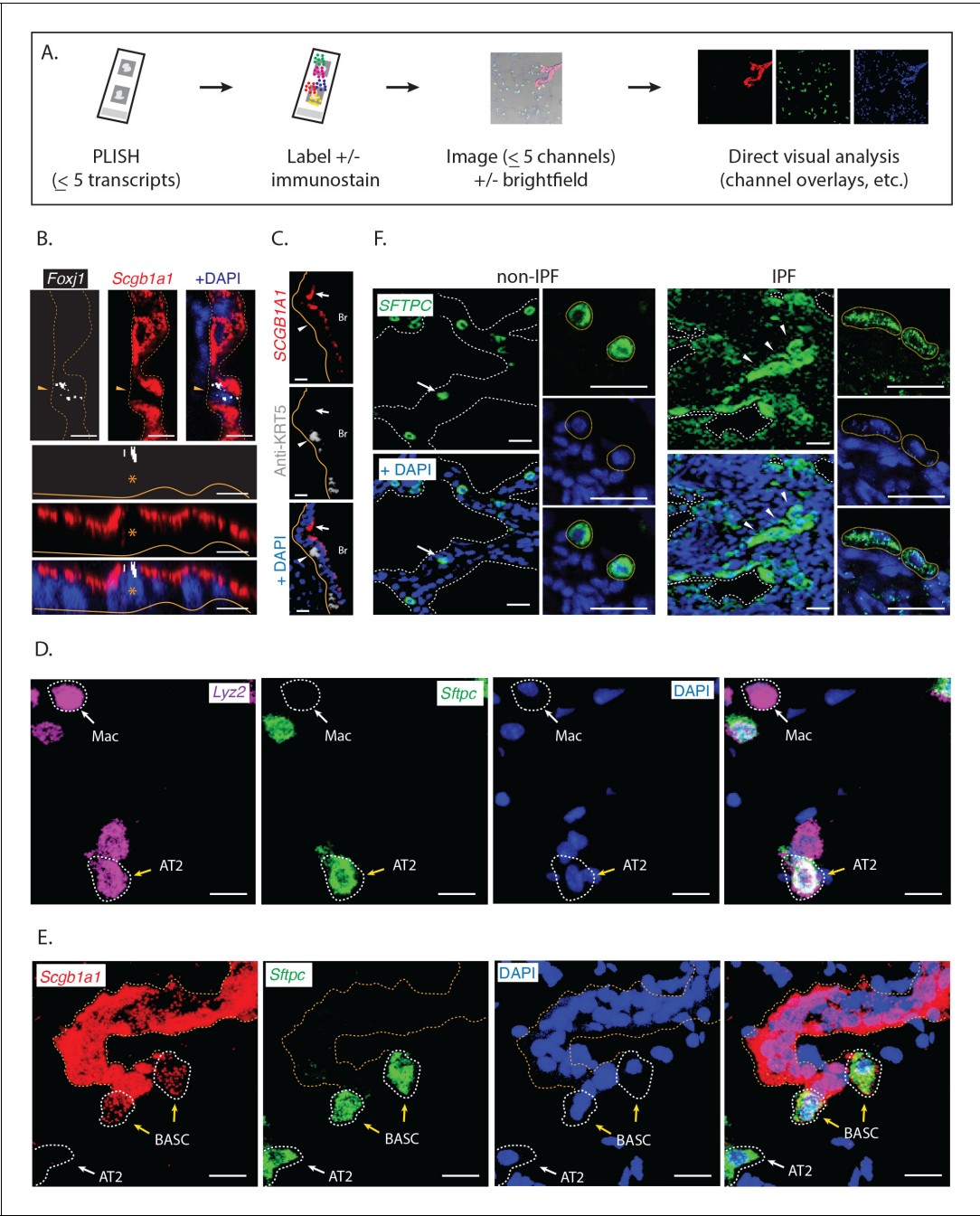

**Figure 2.** Direct visual analysis of single-molecule and single-cell gene expression in diverse specimens. (A) PLISH experimental workflow. After an initial probe hybridization and enzymatic amplification step, up to five distinct channels can be simultaneously detected and imaged by conventional fluorescence microscopy, enabling direct visualization of RNA abundance. (B) PLISH detects single RNA molecules with single-cell resolution in tissues. PLISH staining for *Foxj1* (white) and *Scgb1a1* (red) in the bronchial epithelium of mouse lung shows a single ciliated cell (Foxj1+, arrowhead and asterisk) between Club cells (Scgb1a1+) in a planar view (top) and with orthogonal reconstruction (bottom). Note the discrete white puncta in the ciliated cell, which correspond to single *Foxj1* transcripts. Dashed lines indicate the lateral and solid lines indicate the basal surface of airways. Foxj1, forkhead box J1; Scgb1a1, secretoglobin family 1A member 1; Scale bars, 10 μm. (C) Simultaneous RNA and protein detection in FFPE sections. FFPE human lung co-stained by PLISH (*SCGB1A1*, red) and indirect immunohistochemistry (anti-KRT5, grey) shows the expected localization of Club cells (SCGB1A1+, arrow) and basal cells (KRT5+, arrowhead) along the bronchial (Br) epithelium. Solid lines indicate the basal surface of airways. FFPE, formalin-fixed, paraffin-embedded; KRT5, keratin 5; SCGB1A1, secretoglobin family 1A member 1; Scale bar, 5 μm. (D) Discrimination of AT2 cells from macrophages by visual inspection of RNA abundance. PLISH staining in mouse lung for *Lyz2* (purple) and *Sftpc* (green) allows clear discrimination of alveolar macrophages (Lyz2+ Sftpc−, white arrow) from AT2 cells (Sftpc+ Lyz2+, yellow arrow). AT2, Alveolar epithelial type II; Lyz2, lysozyme 2; Mac, macrophage; Sftpc, surfactant protein C; Scale bar, 20 μm. (E) Discrimination of AT2 cells from BASC cells by visual inspection of RNA abundance.

*Figure 2 continued on next page*

Figure 2 continued

PLISH staining for the Club cell marker (*Scgb1a1*, red) and AT2 cell marker (*Sftpc*, green) shows AT2 (Sftpc⁺), Club (Scgb1a1⁺) and BASC (Sftpc⁺ Scgb1a1$^{Lo}$) cells. Note the discrete red puncta in the 'BASC' cells, which correspond to single *Scgb1a1* transcripts. The cell types localize appropriately, with the AT2 cell in an alveolus (white arrow), the Club cells in the terminal bronchiole, and the double-positive BASCs at the bronchioalveolar junction (yellow arrows). Dashed lines demarcate the airway. AT2, alveolar epithelial type II; BASC, bronchioalveolar stem cell; Scgb1a1, secretoglobin family 1A member 1; Sftpc, surfactant protein C; Scale bar, 20 μm. (F) PLISH in patient tissue samples for molecular analysis of human disease. PLISH staining for *SFTPC* (green) in non-IPF human lung (left) marks AT2 cells (white arrow) distributed within alveolar septae (dashed lines). The adjacent panels show a magnified image of healthy cuboidal AT2 cells (dashed circles). PLISH staining in IPF human lung (right) shows densely cellular regions with architectural distortion of alveolar septae (dashed lines). SFTPC$^{Hi}$ AT2 cells are inappropriately clustered (white arrowheads) and have abnormal flattened morphologies, as seen at higher magnification in right panels (dashed ellipses). AT2, alveolar epithelial type II; IPF, Idiopathic Pulmonary Fibrosis; SFTPC, surfactant protein C; Scale bar, 100 μm.
DOI: https://doi.org/10.7554/eLife.30510.007

## Multiplexed and iterative PLISH in tissues

Highly multiplexed measurement of different RNA species requires iterated data collection cycles, since conventional fluorescence microscopy only provides up to five channels (*Figure 3A*). The data collection cycles include fluorescent labeling of a subset of the 'barcodes' (i.e., unique nucleotide sequences complementary to fluorescently labeled 'imager' oligonucleotides) in a sample, imaging of the labeled transcripts, and erasure of the fluorescent signal. Ideally, the cycles should be fast, and the erasure should not cause any mechanical or chemical damage to the sample. Consistent with prior work, we found that PLISH puncta could be imaged in the presence of a 3 nM background of freely-diffusing imager oligonucleotides (*Figure 3—figure supplements 1A* and [*Blab et al., 2004*]). This allowed us to streamline data collection by eliminating a wash step, and also presented a simple erasure strategy. By using short imager oligonucleotides that equilibrate rapidly on and off of RCA amplicons (*Jungmann et al., 2014*), we could erase fluorescence from a previous cycle by a simple buffer exchange (*Figure 3—figure supplement 1B*). We also established an erasure method based on uracil-containing imager oligonucleotides, which were removed with a 15 min enzymatic digestion (*Figure 3—figure supplement 1C*). Thus, we could image PLISH puncta in five different color channels with spectrally-distinct fluorophores, and we were able to complete cycles in as little as 20 min, which approaches the cycle time of an Illumina MiSeq instrument (https://support.illumina.com).

To demonstrate and validate the multiplexing capacity of PLISH, we co-localized the mRNA of eight selected genes in ~2900 single cells from an adult mouse lung (*Figure 3B*). Our panel included four commonly used lung cell type markers that have been previously characterized, and four ubiquitously expressed genes. The targeted transcripts were *Sftpc* (AT2 cells), advanced glycosylation end product-specific receptor (*Ager*, AT1 cells), *Scgb1a1* (Club cells), *Lyz2* (macrophage and AT2 cell subset), ferritin light polypeptide 1 (*Ftl1*), beta actin (*Actb*), inactive X specific transcripts (*Xist*), and glyceraldehyde-3-phosphate dehydrogenase (*Gapdh*). The eight RNA species were barcoded in a single PLISH reaction, and the data were collected with a pair of label-image-erase cycles using the enzymatic erasure approach described above (*Figure 3A*). A nuclear counterstain (DAPI) and transmitted light micrograph were also obtained.

To quantify the expression of all eight genes on a per-cell basis, we created a PLISH-specific pipeline in CellProfiler, an open-source software package (*Kamentsky et al., 2011*). The pipeline first identified nuclei in the DAPI channel, which were used as anchor points for expansion to full-cell assignments. Fortuitously, the bulk of the detected mRNAs in AT1 cells, which have an extremely flat and broad morphology, were clustered around the nuclei. We summed the PLISH signal for each gene in the nuclear and peri-nuclear regions of each cell, and saved the results as single-cell expression profiles indexed on anatomical location. We also created a utility to pseudocolor cells in a transmitted light micrograph according to their inferred cell type (see below), so that we could visualize the relationship between cellular gene expression and anatomical localization.

## Automated cell classification and insights into lung biology

An important scientific challenge is to identify and map all of the molecularly distinct cell types that make up complex tissues, and in situ single-cell profiling should be a powerful tool for working towards this goal. As a proof-of-concept for this, we asked whether known lung cell types could be

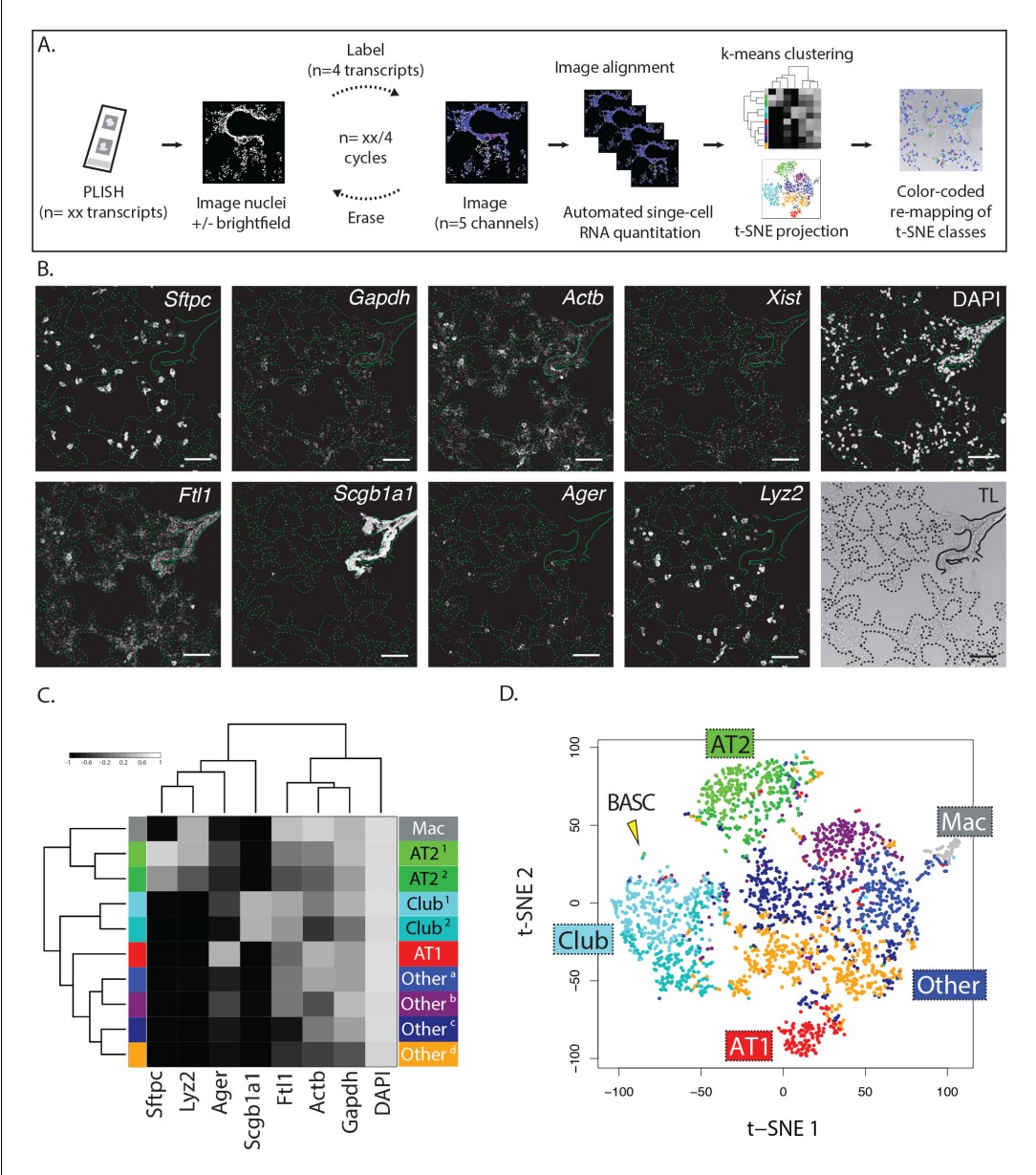

**Figure 3.** Multiplexed PLISH: rapid label-image-erase cycles, automated data analysis, and unsupervised cell classification. (**A**) Multiplexed PLISH experimental workflow. Probes for many different RNAs are hybridized and amplified in a single reaction. The PLISH amplicons marking four RNA species are then labeled with four fluorescent imager oligonucleotides, imaged on a microscope, and 'erased' by elimination of the imager oligonucleotides. Amplicons marking a different subset of four RNAs are then labeled with four new imager oligonucleotides, imaged, and erased. This cycle is repeated until all of the RNA species have been visualized and photo-documented. The images are automatically aligned and processed, and the signal for each RNA species in each cell is summed to produce single-cell expression profiles. Unsupervised k-means clustering of the expression profiles (or other computational tools) empirically identifies distinct cell classes. t-SNE plots (or other data visualization tools) show the differences in gene expression and the classification for all of the cells in a tissue. The location of individual cells or cell classes is spatially remapped onto images of the tissue in order to integrate molecular, histological, and spatial features. (**B**) A multiplexed PLISH data set. Eight different transcripts in mouse lung were visualized with two label-image-erase cycles. A micrograph for each channel in one field of view is shown. Solid lines indicate the basal surface of airways and dashed lines indicate alveolar septae. Actb, beta actin; Ager, advanced glycosylation end product-specific receptor; Ftl1, ferritin light polypeptide 1; Gapdh, glyceraldehyde-3-phosphate dehydrogenase; Lyz2, lysozyme 2; Scgb1a1, secretoglobin family 1A member 1; Sftpc, surfactant protein C; Xist, inactive X specific transcripts. Scale bar, 80 μm. (**C**) Automated cell classification. K-means clustering partitions ~2900 single cells into one of ten molecularly distinct classes, with the expression profile of each cluster centroid displayed in a heat map. Based on marker gene expression, individual clusters were inferred to be macrophages (mac, white), two classes of AT2 cells (light and dark green), two classes of Club cells (light and dark cyan), AT1 cells (red), and four categories of unassigned cell types (shades of blue, purple, and orange). (**D**) An overview of the cells in a murine lung. Differences in gene expression for ~2900 cells are displayed as a two-dimensional t-SNE plot. Each cell is represented by a single dot, colored

*Figure 3 continued on next page*

*Figure 3 continued*

according to its cluster assignment. Labels mark the location of each cell class. The yellow arrowhead indicates a small island of cells that exhibit the profile of BASCs. AT1, alveolar epithelial type I cell; AT2, alveolar epithelial type II cell; BASC, bronchioalveolar stem cell; Mac, macrophage.
DOI: https://doi.org/10.7554/eLife.30510.008
The following figure supplement is available for figure 3:

**Figure supplement 1.** Rapid label-image-erase strategies without tissue degradation.
DOI: https://doi.org/10.7554/eLife.30510.009

rediscovered by an automated and unsupervised analysis of our multiplexed PLISH data set. We used two standard data analysis tools, K-means clustering (*Figure 3C*) and t-distributed stochastic neighbor embedding (*van der Maaten and Hinton, 2008*) (t-SNE, *Figure 3D*), to classify and visualize the entire population of cells. The automated analysis identified ten cell classes, four of which were labeled 'other' because they were defined primarily by 'signature' profiles of ubiquitously-expressed genes. The remaining six classes were associated with a known lung cell type based on marker-gene expression. The *Sftpc* and *Scgb1a1* positive cell classes were labeled as AT2 and Club, respectively, while the *Lyz2* positive class was labeled as macrophage (one of the two AT2 cell classes was also *Lyz2* positive as previously reported in (*Desai et al., 2014*), *Figure 4—figure supplement 1A*). The cell class with the highest *Ager* expression was labeled as AT1, but *Ager* mRNA was also detected in a subset of AT2 and Club cells, and in one of the four 'other' cell classes, indicating it is not particularly specific for AT1 cells. We validated the PLISH results by indirect immunohistochemistry (*Figure 4—figure supplement 1B*) and by comparison with previously published scRNA-seq data (*Figure 4—figure supplement 1C*), which confirmed the low specificity of *Ager* for AT1 cells. We also analyzed the RNA expression of *Akap5*, another transcript that is highly-enriched in AT1 cells (*Treutlein et al., 2014*), and found that its localization correlated closely with *Ager*'s (*Figure 4—figure supplement 1D*).

For a higher-resolution analysis of cellular gene expression, we examined the expression pattern of individual genes in re-colored t-SNE plots (*Figure 4A*). We found a small cloud of cells between the Club and AT2 clusters that expressed both *Sftpc* and *Scgb1a1*. On the basis of this dual expression, we assigned them as the BASC type (*Kim et al., 2005*). We also noted that *Lyz2* expression partitioned the AT2 cells into two classes designated $Lyz2^+$ and $Lyz2^-$, while *Actb* segregated Club cells into two classes designated $Actb^{Hi}$ and $Actb^{Lo}$. *Gapdh* was the most uniformly expressed transcript, consistent with its role as a 'housekeeping' gene (*Figure 4B*). *Ftl1* expression was highest in alveolar macrophages, as expected, where it is believed to play a role in processing iron from ingested red blood cells (*McGowan and Henley, 1988*). Unexpectedly, *Ftl1* was also highly expressed in Club cells. *Actb* expression was highest in macrophages, presumably because of its functional role in motility, and in AT1 cells, which must maintain a flat morphology and expansive cytoskeleton (*Foster et al., 2010*).

To validate the PLISH results, we pseudocolored the cells in transmitted-light images according to their class (*Figure 4C–D*). Importantly, no spatial information was included in the k-means clustering. Several observations confirmed the accuracy of the automated classification. First, the Club cell class mapped perfectly onto the bronchial epithelium, while cells from the AT1 and AT2 classes were distributed throughout the alveolar compartment. The rare BASCs also localized precisely to the bronchioalveolar junctions, where they have been shown to reside by immunostaining (*Kim et al., 2005*) (*Figure 4C* and *Figure 4—figure supplement 1E*). The macrophage class was primarily found inside the alveolar lumen, and many exhibited a characteristic rounded cell shape. The Other$^d$ class of cells was enriched in pulmonary arteries, and therefore might represent endothelial or perivascular cells. We further observed a striking spatial segregation of the two Club cell classes. $Actb^{Hi}$ Club cells clustered together at the bronchial terminus, while $Actb^{Lo}$ Club cells populated more proximal domains (*Figure 4D–E*). While the significance of this pattern is not immediately obvious, it emphasizes how PLISH can readily integrate molecular and spatial features of single cells to generate insights that would be missed with either piece of information alone.

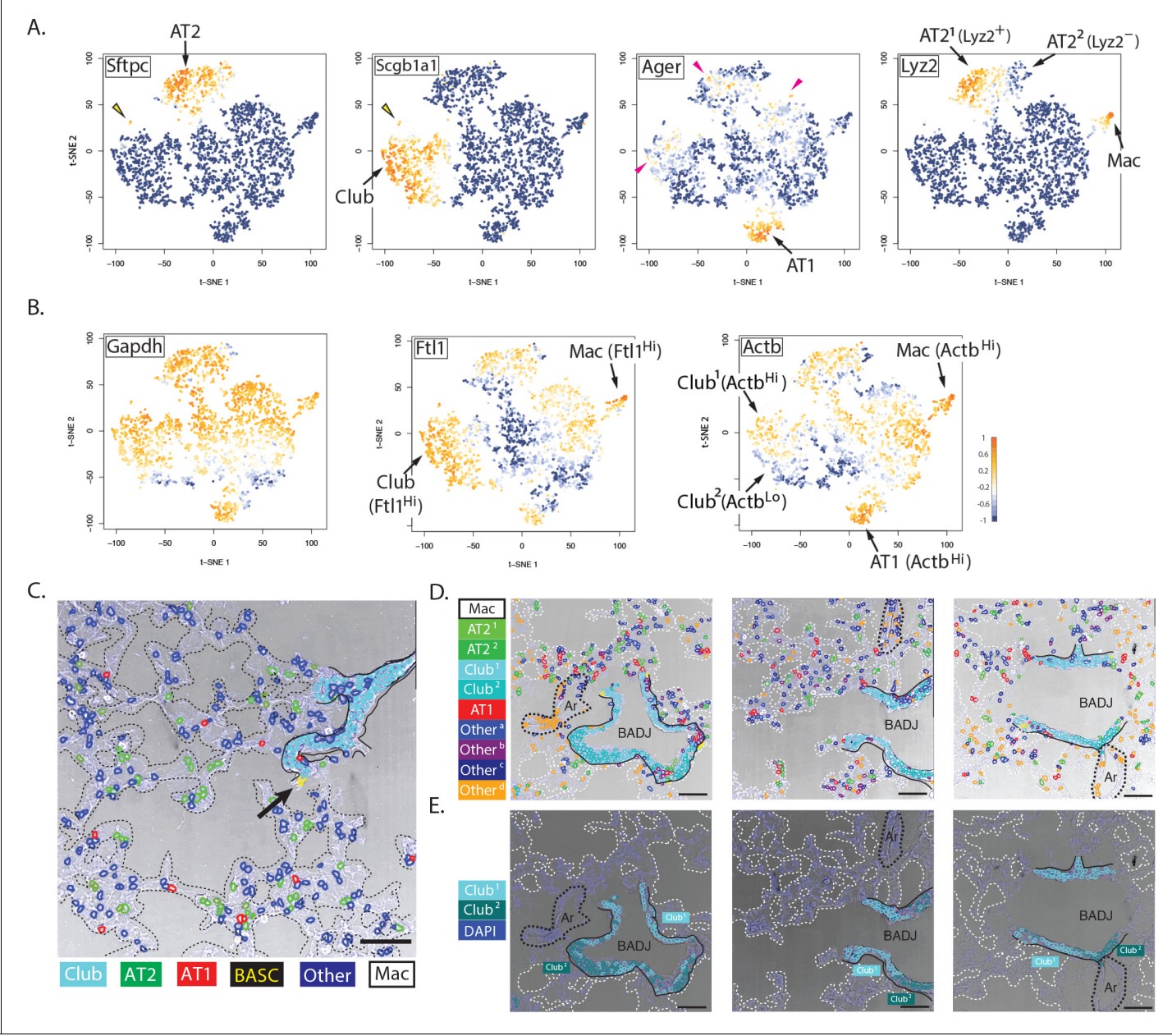

**Figure 4.** Biological insights from integrated molecular and spatial information. (**A**) Specificity and promiscuity in marker gene expression. t-SNE plots are colored according to the expression of four cell-type marker genes. High expression (yellow) in the first two panels highlights AT2 and Club cells, as indicated, while the arrowhead indicates rare double-positive BASCs. The third panel shows high levels of *Ager* in AT1 cells (arrow), but promiscuous expression in a subset of AT2, Club[1] and Other[b] cell classes (red arrowheads). *Lyz2* expression in the fourth panel is restricted to the macrophage and AT2[1] classes. Note that differential *Lyz2* expression splits the canonical AT2 cell type into two sub-classes (AT2[1] and the AT2[2]). (**B**) Differential expression of 'housekeeping' genes in canonical cell types. t-SNE plots are colored according to the expression of three ubiquitous 'housekeeping' genes. *Gapdh* (left) is the most evenly and broadly distributed, while *Ftl1* (middle) is highest in the macrophage and Club cell classes. *Actb* is the highest in the macrophage and AT1 cell classes. Unexpectedly, differential *Actb* expression splits the canonical Club cell type into two sub-classes (Club[1] and Club[2]). (**C**) Spatial organization of lung cell classes. The nuclei of cells in a transmitted light image of a bronchioalveolar duct junction (BADJ) are pseudocolored according to their basic cluster assignment. The Club class (cyan) localizes to the bronchial epithelium, while the AT1 (red) and AT2 (green) cell classes are distributed throughout the alveolar compartment. The macrophage class (white) is primarily found in the alveolar lumen. Rare BASCs (yellow) localize precisely to the bronchioalveolar junctions (arrow), where they have been shown to reside by immunostaining (*Kim et al., 2005*) (*Figure 4C* and *Figure 4—figure supplement 1E*). This image demonstrates how PLISH can be used to localize specific cells of interest within their anatomical context. Solid lines indicate the basal surface of airways and dashed lines demarcate alveolar septae. Scale bar, 80 µm. (**D**) Spatial organization in the terminal airway. The nuclei in three terminal airway fields of view are pseudocolored according to their cluster assignment. Note the presence of both Club[1] and Club[2] cell classes (light and dark cyan), and all four Other cell classes. The Other[d] class (orange) is enriched in pulmonary

*Figure 4 continued on next page*

*Figure 4 continued*

arteries, indicated by black dashed lines, and therefore might represent endothelial or perivascular cells. Solid lines indicate the basal surface of airways, black dashed lines indicate pulmonary arteries, and dashed white lines demarcate alveolar septae. Ar, artery; Scale bar, 80 µm. (**E**) Two sub-classes of Club cells, defined by a difference in *Actb* expression, segregate anatomically. Club cells in the three fields of view from panel D are pseudocolored with enhanced contrast, revealing a striking spatial pattern. The Club[1] sub-class (cyan, Actb$^{Hi}$ Ager$^{Lo}$) localizes to the BADJ, whereas the Club[2] sub-class (blue, Actb$^{Lo}$ Ager$^-$) segregates more proximally in the terminal airways. Differential *Actb* expression might reflect region-specific differences in mechanical stress. The integration of molecular and spatial information in this image reveals biology that would be missed with either piece of information alone. Solid lines indicate the basal surface of airways, black dashed lines indicate pulmonary arteries, and dashed white lines demarcate alveolar septae. Ar, artery; BADJ, bronchioalveolar duct junction; Scale bar, 80 µm.

DOI: https://doi.org/10.7554/eLife.30510.010

The following figure supplement is available for figure 4:

**Figure supplement 1.** Immunohistochemical and single cell RNA-sequencing correlation of PLISH results.

DOI: https://doi.org/10.7554/eLife.30510.011

## Discussion

PLISH represents a practical technology for multiplexed expression profiling in tissues. It combines high performance in four key areas: specificity, detection efficiency, signal-to-noise and speed. The specificity derives from coincidence detection, which requires two probes to hybridize next to one another for signal generation. Efficient detection of low-abundance transcripts is accomplished by targeting multiple sites along the RNA sequence. Enzymatic amplification produces extremely bright puncta, and allows many different RNA transcripts to be marked with unique barcodes in one step. The different RNA transcripts can then be iteratively detected to rapidly generate high dimensional data.

While low-plex PLISH on a handful of different genes can be valuable, the PLISH technology is also scalable, without requiring specialized microscopes (or other equipment), software, or computational expertise. The oligonucleotides and enzymes are inexpensive and commercially available from multiple vendors. The H probes are the cost-limiting reagent, but can be synthesized in pools (*Murgha et al., 2014*; *Beliveau et al., 2012*). Assuming five pairs of H probes for each target RNA species, and 20 cents for a 40mer oligonucleotide, the cost of PLISH reagents amounts to $3 per gene. It should therefore be practical to simultaneously interrogate entire molecular systems, such as signaling pathways or super-families of adhesion receptors. The high specificity and signal-to-noise of PLISH will be advantageous for deep profiling, where non-specific background increases with increasingly complex mixtures of hybridization probes (*Moffitt et al., 2016c*).

Our initial studies demonstrate PLISH's capacity for rapid, automated and unbiased cell-type classification, and illustrate how it can complement single-cell RNA sequencing (sc-RNAseq). Sc-RNAseq offers greater gene depth than in situ hybridization approaches, but it is less sensitive, fails to capture spatial information, and induces artefactual changes in gene expression during tissue dissociation (*van den Brink et al., 2017*; *Lee et al., 2015*). PLISH provides the missing cytological and spatial information, and it is applied to intact tissues. Going forward, sequencing can be used to nominate putative cell types and molecular states based on the coordinate expression of 'signature genes', and multiplexed PLISH can be used to distinguish true biological variation from technical noise and experimentally-induced perturbations. Importantly, multiplexed PLISH provides the tissue context of distinct cell populations, which is essential for understanding the higher-order organization of intact systems like solid tumors and developing organs. In diseases like IPF where morphology and gene expression are severely deranged (*Xu et al., 2016*), histological, cytological and spatial features may even be essential for making biological sense of sequencing data.

Currently, efforts are underway to more deeply characterize cellular states by integrating diverse types of molecular information. We have already demonstrated the combined application of PLISH with conventional immunostaining. Going one step further, oligonucleotide-antibody conjugates make it possible to mix and match protein and RNA targets in a multiplexed format (*Weibrecht et al., 2013*). The generation of comprehensive, multidimensional molecular maps of intact tissues, in both healthy and diseased states, will have a fundamental impact on basic science and medicine.

# Materials and methods

## Materials

Unless otherwise specified, all reagents were from Thermo-Fisher and Sigma-Aldrich. Oligonucleotides were purchased from Integrated DNA Technologies. T4 polynucleotide kinase, T4 ligase, USER enzyme and their respective buffers were purchased from New England Biolabs. Nxgen phi29 polymerase and its buffer were purchased from Lucigen.

Abbreviations: BSA, bovine serum albumin; DAPI, 4,6-diamidino-2-phenylindole; DEPC, diethylpyrocarbonate; EDTA, ethylenediaminetetraacetic acid; min, minutes; PBS, phosphate buffered saline; PFA, paraformaldehyde; RCA, rolling circle amplification; RT, room temperature. All oligonucleotide sequences are listed in *Supplementary file 1*.

## Sample preparation

HCT116 cells (ATCC; CCL-247) were authenticated by HLA typing and confirmed negative for *Mycoplasma* contamination using PCR. Cells were grown on poly-lysine coated #1.5 coverslips (Fisherbrand 12–544 G) using standard cell culture protocols until they reached the desired confluency. The cells were rinsed in 1X PBS and fixed in 3.7% formaldehyde with 0.1% DEPC at RT for 20 min. The fixed cells were treated with 10 mM citrate buffer (pH 6.0) at 70°C for 30 min, dehydrated in an ethanol series, then enclosed by application of a seal chamber (Grace Biolabs 621505) to the coverslip.

Lungs were collected from adult B6 mice (Jackson Labs) and fixed by immersion in 4% PFA as previously described (*Desai et al., 2014*). Non-IPF human lung tissue was obtained from a surgical resection, and IPF tissue from an explant. All mouse and human research were approved by the Institutional Animal Care and Use Committee and Internal Review Board, respectively, at Stanford University. The tissues were fixed by immersion in 10% neutral buffered formalin in PBS at 4°C overnight under gentle rocking, cryoprotected in 30% sucrose at 4°C overnight, submerged in OCT (Tissue Tek) in an embedding mold, frozen on dry ice, and stored at −80°C. 20 μm sections were cut on a cryostat (LeicaCM 3050S) and collected on either poly-lysine coated #1.5 coverslips or glass slides (Fisherbrand Superfrost), air dried for 10 min, and post-fixed with 4% PFA at RT for 20 min. The human lung tissue in *Figure 2A* was formalin-fixed and paraffin-embedded (FFPE) according to standard protocols, and 20 μm sections were cut on a microtome and collected on glass slides. The FFPE sections were deparaffinized by immersion in Histoclear (National Diagnostics, HS-200) for 3 × 5 min, then dehydrated in an ethanol series and post-fixed with 4% PFA at RT for 20 min. Tissue sections were treated with 10 mM citrate buffer (pH 6.0) containing 0.05% lithium dodecyl sulfate at 70°C for 30 min, or in some experiments, with 0.1 mg/ml Pepsin in 0.1M HCl for 8 min at 37°C and dehydrated in an ethanol series. Following treatment, sections were air dried for 10 min and enclosed by application of a seal chamber.

## PLISH probe design and preparation

Target RNAs were probed at ~40 nucleotide detection sites, with 1 to 10 sites per RNA species depending on expression level. NCBI BLAST searches were used to eliminate detection sites that shared 10 or more contiguous nucleotides with a non-target RNA. The detection sites were also selected to minimize self-complementarity as indicated by the IDT oligo analyzer. Each detection site was targeted with a pair of H probes designated HL (left H probe) and HR (right H probe). The HL and HR probes included ~20 nucleotide binding sequences that were complementary respectively to the 5' and 3' halves of the detection site. The binding sequences were chosen so that the 5' end of the HL binding sequence and the 3' end of the HR binding sequence would abut at a 5'-AG-3' or a 5'-TA-3' dinucleotide in the target RNA. The lengths of the binding sequences were adjusted so that the melting temperature of the corresponding DNA duplex would fall between 45–65°C as computed by IDT Oligo analyzer using default settings of 0.25 μM oligo concentration and 50 mM salt concentration. To generate H probes, suitable HL and HR binding sequences were catenated at their respective 5' and 3' ends with overhang sequences taken from one of eight modular design templates (*Supplementary file 1*). The left and right overhang sequences in each design template were complementary to a specific bridge (B) and circle (C) oligonucleotide, which directed a desired fluorescent readout. The design templates reported here utilized a common 31 base oligonucleotide for the bridge. Following previous work (*Söderberg et al., 2006*), the circle oligonucleotides

were ~60 bases long with 11 base regions of complementarity to cognate H probes on either end. The circle sequences were chosen to minimize self-complementarity. Each imager oligonucleotide was complementary to a barcode embedded in one of the C oligonucleotides, allowing unique detection of the corresponding RCA amplicon.

The H-probe oligonucleotides were ordered on a 25 nanomole scale with standard desalting. The B and C oligonucleotides were ordered on a 100 nanomole scale with HPLC purification, and phosphorylated with T4 polynucleotide kinase according to the manufacturer recommendations. Imager oligonucleotides were purchased either as HPLC-purified fluorophore conjugates (A488, Texas Red, Cy3, Cy5), or as amine-modified oligonucleotides that were subsequently coupled to Pacific Blue-NHS ester according to the manufacturer recommendations.

## PLISH barcoding procedure

Six buffers were used for PLISH barcoding: H-probe buffer (1M sodium trichloroacetate, 50 mM Tris pH 7.4, 5 mM EDTA, 0.2 mg/mL Heparin), bridge-circle buffer (2% BSA, 0.2 mg/mL heparin, 0.05% Tween-20, 1X T4 ligase buffer in RNAse-free water), PBST (PBS + 0.1% Tween-20), ligation buffer (10 CEU/µl T4 DNA ligase, 2% BSA, 1X T4 ligase buffer, 1% RNaseOUT and 0.05% Tween-20 in RNAse-free water), labeling buffer (2x SSC/20% formamide in RNAse-free water), and RCA buffer (1 U/µl Nxgen phi29 polymerase, 1X Nxgen phi29 polymerase buffer, 2%BSA, 5% glycerol, 10 mM dNTPs, 1% RNaseOUT in RNAse-free water).

An H cocktail was prepared by mixing H probes in H-probe buffer at a final concentration of 100 nM each. If an RNA was targeted with more than five probe sets, the concentrations of the H probes for that RNA were pro-rated so that their sum did not exceed 1000 nM. A BC cocktail was also prepared by mixing B and C oligonucleotides in bridge-circle buffer at a final concentration of 6 µM each.

Single-step barcoding was performed in sealed chambers. The workflow consisted of three steps: (i) The sample was incubated in the H cocktail at 37°C for 2 hr. The sample was then washed 4 × 5 min with H-probe buffer at RT, and incubated in the BC cocktail at 37°C for 1 hr. (ii) Following a 5 min wash with PBST at RT, the sample was incubated in ligation buffer at 37°C for 1 hr. (iii) The sample was washed 2 × 5 min with labeling buffer at RT, and washed with 1X Nxgen phi29 polymerase buffer at RT for 5 min. The sample was then incubated in RCA buffer at 37°C for 2 hr (typical for cultured cells) to overnight (typical for tissue). Finally, the sample was washed 2 × 5 min with labeling buffer.

## Imaging

Barcoded PLISH samples were fluorescently labeled by two different procedures, designated 'washout' and 'fast'. In the washout procedure, the sample was incubated with imager oligonucleotides in imager buffer (labeling buffer with 0.2 mg/mL heparin) at a final concentration of 100 nM each for 30 min, and then washed 2 × 5 min with PBST at RT. In the fast procedure, the sample was incubated for 5 min with imager oligonucleotides in imager buffer at a final concentration of 3 nM each, and then imaged immediately. Samples that did not require label-image-erase cycles were stained with DAPI (stock 1 mg/ml; final concentration - 1:1000 in PBS) for 5 min and mounted in H-1000 Vectashield mounting medium (Vector).

Data were collected by confocal microscopy (Leica Sp8 and Zeiss LSM 800) using a 40X oil immersion or a 25X water immersion objective lens. 20 µm z-stacks were scanned, and maximum projection images were saved for analysis. For 5-color experiments, DAPI was added after the Pacific Blue channel had been imaged, and the Texas Red and Cy3 channels were linearly unmixed using Zeiss software. Transmitted light images were acquired on a Leica Sp8 confocal microscope using the 488 nm Argon laser and the appropriate PMT-TL detector. Images from serial rounds of data collection were aligned using the nuclear stain from each round as a fiducial marker. Unless otherwise stated, imaging data of cells and mouse lung tissue are representative of three independent experiments with ≥4 fields of view each. Imaging data of human lung tissue are representative of two independent experiments with ≥4 fields of view each.

## PLISH and HCR co-localization

HCR was performed following a published protocol (*Choi et al., 2014*) with probes that targeted two sites covering nucleotides 621–670 and 1159–1208 in the mouse Axin2 transcript, and Alexa-Fluor 488-/AlexaFluor 647-labeled amplifier oligonucleotides. The samples were then processed for PLISH with H probes targeting four sites covering nucleotides 347–386, 1878–1917, 2412–2451 and 2956–2995 in the Axin2 transcript, and imaged using a Cy3-labeled imager oligonucleotide.

## PLISH with concurrent immunohistochemistry

PLISH barcoding was performed as described above. Subsequently, the sample was washed 3 × 5 min with PBST at RT, and incubated in blocking solution (50 μl/ml [5%] normal goat serum, 1 μl/ml [0.1%] Triton X-100, 5 mM EDTA and 0.03 g/ml [3%] BSA in PBS) at RT for 1 hr. The sample was then incubated with primary antibody (Rabbit anti-pro-Sftpc, Millipore, 1:500 or Rabbit anti-Cytokeratin 5, Abcam Ab193895, 1:400) in blocking solution at 37°C for 2 hr under gentle rocking, washed 4 × 5 min with PBST at RT, and incubated with secondary antibody (Goat anti-Rabbit-Cy5, Jackson Lab, 1:250) and DAPI (1:1000) in blocking solution at RT for 1 hr. The sample was washed 3 × 5 min in PBST at RT and mounted in H-1000 Vectashield.

## Antisense blocking oligonucleotide

Mouse lung tissue cryosections were collected on slides, post-fixed and processed as described above. The samples were incubated with a 60-base oligonucleotide complementary to nucleotides 219–278 in the Scgb1a1 mRNA, or with a scrambled 60-base oligonucleotide, at 100 nM final concentration in H-probe buffer at 37°C for 2 hr. The samples were then washed 2 × 5 min with H-probe buffer at RT, and processed for PLISH using H probes that targeted nucleotides 229–268 in the Scgb1a1 transcript.

## Signal erasure for iterative cycles of PLISH

To perform enzymatic erasure, 15–20 base imager oligonucleotides were ordered with the dT nucleotides replaced by dU nucleotides. Following imaging, the signal was erased by incubating the sample with 0.1 U/μL USER enzyme in 1X USER enzyme buffer at 37°C for 20 min, followed by washing 2 × 3 min with PBST at RT. To perform rapid erasure, short 10–11 base oligonucleotides were ordered. Following imaging, the signal was erased by incubating the sample with PBST at 37°C for 15 min.

## Correlative immunostaining

Lungs collected from B6 and the Lyz2$^{+/EGFP}$ mouse strains (*Faust et al., 2000*) were fixed and immunostained as whole mounts as previously described (*Desai et al., 2014*). Primary antibodies were chicken anti-GFP (Abcam ab13970), rat anti-Ecad/Cdh1 (Invitrogen ECCD-2), goat anti-Scgb1a1 (gift from Barry Stripp), rabbit anti-pro-Sftpc (Chemicon AB3786), and rat anti-Ager (R and D MAB1179). Fluorophore-conjugated secondary antibodies raised in Goat (Invitrogen) or Donkey (Jackson Labs) were used at 1:250 and DAPI at 1:1000.

## Data analysis

FIJI was used to pseudocolor unprocessed micrographs for display as three-color overlays. A custom CellProfiler (*Kamentsky et al., 2011*) pipeline (*Source code 1*) was created to measure RNA signal intensities at the single-cell level. Briefly, the centers of cell nuclei were first identified as maxima in a filtered DAPI image, and associated with a numerical index. Nuclear boundaries were assigned by a propagation algorithm, and then expanded by ~1 micron to define sampling areas. The following data were then recorded: (i) average pixel intensities for each data channel over each sampling area; (ii) the coordinates of the sampling areas; (iii) shape metrics for the corresponding nuclei; and (iv) an image with the boundary pixels of each nucleus set equal to the associated index value. For each RNA species, the PLISH data were first normalized onto a 0:10 scale by dividing through by the largest value observed in any cell over all of the fields of view, and then multiplying by ten. The data were then log-transformed onto a −1:1 scale by the operation: transformed_data = log(0.1 + normalized_data). Custom Matlab scripts were used to perform hierarchical clustering of the log-transformed single-cell expression profiles, to generate heatmaps, and to create images with the

boundary pixels of each nucleus colored according to a cluster assignment (*Source code 1*). Custom R scripts were used for k-means clustering and to make t-SNE projection plots (*Source code 1*).

## Acknowledgements

We thank Andres Andalon and Yana Kazadaeva for technical assistance; Barry Stripp for the goat anti-Scgb1a1 antibody. This work was supported by the NHLBI Progenitor Cell Biology Consortium 5U01HL09999507 (PH, TD), NHLBI 1R56HL1274701 (TD), Stanford BIO-X IIP-130 (PH, TD), Stanford ChEM-H (MN, PH, TD), and by a Stanford Discovery Innovation Fund award (PH).

## Additional information

### Competing interests

Monica Nagendran: MN has filed a provisional patent for PLISH (Application #62/475,090). Daniel P Riordan: DR has filed a provisional patent for PLISH (Application #62/475,090). Pehr B Harbury: PH has filed a provisional patent for PLISH (Application #62/475,090). Tushar J Desai: TD has filed a provisional patent for PLISH (Application #62/475,090).

### Funding

| Funder | Grant reference number | Author |
| --- | --- | --- |
| National Heart, Lung, and Blood Institute | 5U01HL09999507 | Pehr B Harbury Tushar J Desai |
| National Heart, Lung, and Blood Institute | 1R56HL1274701 | Tushar J Desai |
| Stanford University School of Medicine | BIO-X IIP-130 | Pehr B Harbury Tushar J Desai |
| Stanford University School of Medicine | ChEM-H | Monica Nagendran Pehr B Harbury Tushar J Desai |
| Stanford University School of Medicine | Discovery Innovation Fund Award | Pehr B Harbury |

The funders had no role in study design, data collection and interpretation, or the decision to submit the work for publication.

### Author contributions

Monica Nagendran, Resources, Data curation, Formal analysis, Funding acquisition, Validation, Investigation, Methodology, Writing—original draft; Daniel P Riordan, Conceptualization, Data curation, Formal analysis, Methodology; Pehr B Harbury, Conceptualization, Formal analysis, Supervision, Funding acquisition, Investigation, Methodology, Writing—original draft, Project administration, Writing—review and editing; Tushar J Desai, Conceptualization, Resources, Formal analysis, Supervision, Funding acquisition, Investigation, Methodology, Writing—original draft, Project administration, Writing—review and editing

### Author ORCIDs

Pehr B Harbury https://orcid.org/0000-0002-3657-5485
Tushar J Desai https://orcid.org/0000-0002-8794-5319

### Ethics

Human subjects: Adult human lung was obtained from Stanford Healthcare with patient informed consent and consent to publish in strict accordance with protocol 18891, approved by the Institutional Review Board Administrative Panel on Human Subjects in Medical Research of Stanford University, in compliance with requirements for protection of human subjects.

Animal experimentation: This study was performed in strict accordance with the recommendations in the Guide for the Care and Use of Laboratory Animals of the National Institutes of Health. All of the animals were handled according to approved institutional animal care and use committee (IACUC) protocols (#22988) of Stanford University. The protocol was approved by the Administrative Panel on Laboratory Animal Care (APLAC) of Stanford University. Every effort was made to minimize suffering.

### Decision letter and Author response
Decision letter https://doi.org/10.7554/eLife.30510.018
Author response https://doi.org/10.7554/eLife.30510.019

## Additional files

### Supplementary files
• Source code 1. Custom scripts.
DOI: https://doi.org/10.7554/eLife.30510.012

• Supplementary file 1. Oligonucleotide sequences.
DOI: https://doi.org/10.7554/eLife.30510.013

• Transparent reporting form
DOI: https://doi.org/10.7554/eLife.30510.014

### Major datasets
The following previously published dataset was used:

| Author(s) | Year | Dataset title | Dataset URL | Database, license, and accessibility information |
|---|---|---|---|---|
| Treutlein B, Brownfield DG, Wu AR, Neff NF, Mantalas GL, Espinoza FH, Desai TJ, Krasnow MA, Quake SR | 2014 | Reconstructing lineage hierarchies of the distal lung epithelium using single-cell RNA-seq | https://www.ncbi.nlm.nih.gov/geo/query/acc.cgi?acc=GSE52583 | Publicly available at the NCBI Gene Expression Omnibus (accession no: GSE52583) |

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
