## [Decision Letter]

[Editors’ note: this article was originally rejected after discussions between the reviewers, but the authors were invited to resubmit after an appeal against the decision.]

Thank you for submitting your work entitled "Automated cell type classification in intact tissues by single-cell molecular profiling" for consideration by *eLife*. Your article has been evaluated by a Senior Editor and three reviewers, one of whom is a member of our Board of Reviewing Editors. The reviewers have opted to remain anonymous.

Our decision has been reached after consultation between the reviewers. Based on these discussions and the individual reviews below, we regret to inform you that your work will not be considered further for publication in *eLife*.

While all three reviewers agree that the techniques presented in this manuscript are valuable to the community, all three note there is no substantive biological advance in terms of new knowledge. Furthermore, the imaging data requires improvement as below and low-level transcript detection might add to this paper that should be published in a methods journal. Additionally, novel markers and their expression pattern should be described even in a methods paper.

*Reviewer #1:*

In this study, Nagendran et al. validate the use of an improved version of RNA in situ hybridization that they name PLISH, which utilizes rolling circle amplification rather than the proprietary amplification procedures commercially available (i.e. RNAScope, ViewRNA, etc.), thus providing an accessible tool for the broad scientific community. They demonstrate sensitivity and specificity of PLISH, as well as an application in tissue as opposed to only cultured cells.

However, they make many claims that are not supported by the data and are not particular improvements on previous technology. For example, PLISH puncta in tissues are not on par with IF, and do not offer any subcellular resolution. They go on to use a clever approach to multiplexing and image analysis that overly complicates the presentation of a few data points attempting to mimic the computational approaches necessary for scRNASeq data. The novelty suggested in the cover letter with regard to new cell types is not reflected in the manuscript or the data.

This work is much better suited to a methods journal, as there is no exploratory work, and only validation and confirmation of what was already known. Overall, PLISH seems to be a poor-man's RNAScope/ViewRNA/etc… without the commercialization and therefore cost. While this is a great thing, some of the technical terms thrown around seem to be an attempt to differentiate PLISH from technologies upon which it is only building, not pioneering.

1) Pixels in tissue images look highly saturated/blurry, even DAPI – why?

-Does not help support claims of superior resolution/is not subcellular.

2) PLISH gives puncta, as described, in cells, but looks like weak IF in tissues with no clear subcellular resolution (which is a major claim in the Introduction).

3) Scale bars need to be in every separate image, especially in diseased vs. normal e.g. Figure 2.

4) Dotted lines outlining relative structures, such as basement membrane/lumen, would make it easier to follow.

5) Define "technical terms" if this is to be "biologist friendly" – "imager oligonucleotide" is not clear without a search in the Materials and methods. Workflow in Figure 4 is not readily apparent or obvious. As a major aspect of PLISH, this would be easier appreciated if described more thoroughly.

6) Usage of terms like "barcode" and "imager nt" make something that is not complex much more complicated and "jargon-y" than necessary. "RCA followed by hybridization of a fluorescently labeled nt probe" would be much simpler.

7) The inclusion of number of cells (2900) and visualization of the data in tSNE plots deceivingly equates PLISH to a sequencing approach, rather than a standard snapshot visualization tool.

8) Higher mag images are needed for Figure 3 to support the claim that this is "on par with IF" – the current images are not on par with IF and hardly support the presence of a putative BASC, while the IF in Figure 4—figure supplement 2E is much clearer.

9) Figure 3DE – what is the "other"? How does is fit in with what we already know, since these are a set of well-characterized genes? Should rework the text/figure in 3D to better lead into the application of this aspect of PLISH in Figure 4.

General comments:

Overall, this is a highly technical paper that defines a tool. The authors clearly realize this, as they have filed a patent on PLISH. The concept is interesting and its application has great potential, yet it is not as excitingly novel as the text implies. While indeed an improvement on current technologies, this is fit for a methods report rather than a scientific article. The data are confirmatory rather than exploratory, and serve mainly as tool validation.

The focus on BASCs does not help the authors' case, as these are a controversial cell type not found in humans, and the IF images were more convincing than the PLISH data.

If the authors tackled more profound and subtle questions with PLISH, and in human samples (such as identifying the human BASC), then this would be less a tools paper and more a scientific article.

Finally, the overall tone describing this technology is overly ambitious and exceedingly strong. While this is fantastic work at method development and optimization (as well as creating accessibility), it falls short of being an original research article. The cover letter overstates the novelty.

*Reviewer #2:*

This manuscript tackles an important question: how to obtain spatial information about RNA expression in a high-throughput fashion. It presents a technique the authors call PLISH which is an improved variation of the "in situ sequencing" technique published in 2013. The manuscript convincingly demonstrates that in cultured cells the PLISH technique can detect a variety of transcripts which are expressed at different levels and that PLISH detection levels correlate extremely well with FPKM values obtained by RNA-seq. In addition, they provide proof-of-principle that in tissue sections their technique can be used in a multiplex fashion to detect highly-expressed transcripts with very good specificity e.g. the blocking of the Scgb1a1 signal by antisense oligos, but not scrambled oligos, in Figure 1 is very impressive given how highly Scgb1a1 is transcribed.

Beyond Figure 1, the additional experiments presented all show that the PLISH technique works well in mouse and human lung sections (which are complex tissues) when highly-expressed mRNAs are detected.

However, no biological insights are presented. For example, Figure 2 shows that the technique basically works in formalin-fized paraffin embedded human samples using extremely highly expressed transcripts, but these results had previously been obtained by immunostaining so no biological insights are presented. Note: the cell shape changes that are described in the text for SpC+ cells are not visible in the image on the merged pdf.

The analysis of adult mouse lungs – Figure 3 and Figure 4 is a bit disappointing. This is solid proof-of-principle that the PLISH technique works. However, these transcripts are all highly expressed – no real biological insights are gained at all here. To publish in *eLife* I would at least expect to see low level transcripts detected, or some new biology, in addition to the proof of principal work shown here. No prior evidence springs to mind as to what the 2 types of club cells could be – prox and dist, but this change in B-actin could be purely structural to do with cell size/shape in different parts of the airways.

"Proper" analysis of a tissue section using probes to transcripts that are thought to be expressed at low levels, in addition to highly expressed genes such as Sftpc, is required. Plus some biological insight.

*Reviewer #3:*

The present study by the Desai lab shows the development of a proximity ligation in situ hybridization technology and its use to examine various cell type markers in the mouse and human lung. The technique appears to work well and if employed correctly could enhance studies on human tissue samples where histology is possible but isolation of single cells is limited. One limitation of the current study is that it does not extend our understanding of lung cell heterogeneity as most if not all of the markers examined have already been extensively characterized in the lung.

1) Much of the histology shown is at far too low of a magnification to assess the patterns of expression. The authors should provide high mag pictures in all situations, especially for the alveolar region as it is difficult to assess where the positive cells are located. Use of 3D reconvolution (i.e. IMARIS) would be helpful.

2) It would be nice for the authors to show data from some new novel markers that were isolated from the current study. This would provide novel insight into how this technique has advantages over other techniques.

3) A lot of the co-staining is difficult to assess. These data may be helped by higher mag imaging.

4) In the scRNA-seq experiment, was this done on whole lung cell suspension or Epcam+? It is not mentioned in the paper that I can see.

5) Many of the figures need better annotation. It’s very hard to interpret what some of the panels are trying to convey.

---

## [Author Response]

[Editors’ note: the author responses to the first round of peer review follow.]

Reviewer #1:In this study, Nagendran et al. validate the use of an improved version of RNA in situ hybridization that they name PLISH, which utilizes rolling circle amplification rather than the proprietary amplification procedures commercially available (i.e. RNAScope, ViewRNA, etc.), thus providing an accessible tool for the broad scientific community. They demonstrate sensitivity and specificity of PLISH, as well as an application in tissue as opposed to only cultured cells.However, they make many claims that are not supported by the data and are not particular improvements on previous technology. For example, PLISH puncta in tissues are not on par with IF, and do not offer any subcellular resolution.

By "on par with fluorescence immunostaining" we meant that the differences in single-cell gene expression can be visualized in PLISH micrographs without computational processing (better illustrated now in the high-magnification panels of Figure 2). We have clarified our language on this point. PLISH's exceptional signal-to-noise is a striking improvement over tiling and HCR approaches, which because they require computational processing to remove false positives and to improve signal-to-noise, are not amenable to direct visual analysis.

With respect to 'subcellular resolution', we use the term in an introductory paragraph describing past applications of single-molecule in situ hybridization (smISH). We did not make any claims about the subcellular resolution of PLISH in tissue. However, PLISH puncta can in fact be localized with subcellular resolution. The revised high-magnification micrograph of Scgb1a1 RNA in BASC cells (Figure 2) now illustrates this point, as does the micrograph of Foxj1 RNA in ciliated cells (Figure 2). Although the ability to resolve single RNA molecules will be useful for studies of subcellular RNA distribution in tissues, this was not an objective of our work.

It is not clear to us what other claims the reviewer feels are unsupported. A more detailed explanation would be helpful.

They go on to use a clever approach to multiplexing and image analysis that overly complicates the presentation of a few data points attempting to mimic the computational approaches necessary for scRNASeq data.

K-means clustering and tSNE projections are standard tools for reducing the complexity of high-dimensional data sets. We applied them to our multiplexed PLISH data so that it would be comprehensible to us, and hopefully to readers. Without data reduction, we could not have analyzed the coordinate expression levels of 8 transcripts in 2900 cells (which led to the discovery of systematic differences in housekeeping gene expression between sub-classes).

The novelty suggested in the cover letter with regard to new cell types is not reflected in the manuscript or the data.

The cover letter states that we discovered "two spatially segregated and transcriptionally distinct sub-classes of an airway cell type", which is a factual description of the data. We do not believe that the Club cell sub-classes represent "new cell types", but more likely that the difference in β actin expression between them relates to regional differences in mechanical forces within the terminal bronchioles, as suggested by reviewer #2.

This work is much better suited to a methods journal, as there is no exploratory work, and only validation and confirmation of what was already known. Overall, PLISH seems to be a poor-man's RNAScope/ViewRNA/etc. without the commercialization and therefore cost. While this is a great thing, some of the technical terms thrown around seem to be an attempt to differentiate PLISH from technologies upon which it is only building, not pioneering.

We agree that our manuscript could be viewed as a technical report, so we restructured it as a Tools and Resourcesarticle.

No available smISH technology (including RNAScope, ViewRNA, Stellaris, HCR etc.) is capable of fast expression profiling (>5 genes) in tissue. PLISH is a pioneering technological advance on this basis alone.

Our data show that even modest multiplexing (8 transcripts) for cell type markers and 'housekeeping' genes can reveal new biology. Our observations include: (i) that Ager is not an AT1-specific marker (Figure 3, Figure 4, Figure 4—figure supplement 1), (ii) that Lyz2 expression distinguishes a subset of AT2 cells (Figure 2, Figure 3, Figure 4, Figure 4—figure supplement 1), and (iii) that β-Actin levels distinguish terminal versus pre-terminal airway Club cells (Figure 3, Figure 4).

The technical basis for PLISH is proximity ligation and rolling circle amplification, which has nothing in common with RNAScope, ViewRNA, Stellaris or HCR. PLISH's enzymatic amplification is a departure from the DNA-based amplification used by these alternate methods, and therefore cannot be described as building upon them.

1) Pixels in tissue images look highly saturated/blurry, even DAPI – why?-Does not help support claims of superior resolution/is not subcellular.

We apologize for the low quality of the micrographs, which suffered from file compression and inadequate magnification. We have substituted or added higher-magnification insets to the figures showing raw PLISH micrographs, and we have corrected the compression problem.

Regarding superior or subcellular resolution, please see above. To summarize: (i) we made no claims about superior or subcellular resolution; (ii) nonetheless, we have modified Figure 2 to illustrate subcellular resolution in tissue; (iii) however, our objective in this manuscript was to show that PLISH can identify, localize, and quantify molecularly distinct cell classes. This does not require subcellular localization of PLISH puncta.

2) PLISH gives puncta, as described, in cells, but looks like weak IF in tissues with no clear subcellular resolution (which is a major claim in the Introduction).

We used the term 'subcellular resolution' only once in the manuscript, in an introductory paragraph describing past applications of smISH. We did not intend for this to be perceived as a major claim of our manuscript.

We now show that PLISH can precisely localize transcripts within cells (detailed above). Because the RNAs we selected to profile do not have unusual subcellular distributions, we have not emphasized this feature.

3) Scale bars need to be in every separate image, especially in diseased vs. normal e.g. Figure 2.

We thank the reviewer for catching this, and have added the scale bars as requested.

4) Dotted lines outlining relative structures, such as basement membrane/lumen, would make it easier to follow.

We have now included dotted outlines and increased the magnification of the micrographs to improve clarity.

5) Define "technical terms" if this is to be "biologist friendly" – "imager oligonucleotide" is not clear without a search in the Materials and methods. Workflow in Figure 4 is not readily apparent or obvious. As a major aspect of PLISH, this would be easier appreciated if described more thoroughly.

We completely agree. We have significantly re-organized the manuscript to make the technical aspects easier to follow, including the addition of two workflow schematics (one for simple PLISH and the second for multiplexed PLISH with automated analysis). We also now define technical terms (including "imager oligonucleotides") where they first appear in the figure legends.

We regret the confusion caused by our use of "biologist friendly". We meant that performing the staining does not require any special molecular biology or computational expertise. We have clarified the language in the text on this point.

6) Usage of terms like "barcode" and "imager nt" make something that is not complex much more complicated and "jargon-y" than necessary. "RCA followed by hybridization of a fluorescently labeled nt probe" would be much simpler.

We have modified the text to minimize the use of 'jargon-y' language. In sections of the manuscript focused on multiplexed PLISH, however, we still use the term "barcodes" to connote a set of recognition sequences with minimal cross hybridization. We also still use the term "imager oligonucleotide" rather than "fluorescently labeled probe" to verbally differentiate between the RNA-detection probes (the "H" probes) and the oligonucleotides that toggle puncta on/off during label-image-erase cycles.

7) The inclusion of number of cells (2900) and visualization of the data in tSNE plots deceivingly equates PLISH to a sequencing approach, rather than a standard snapshot visualization tool.

We are puzzled by the assertion that we are 'deceiving' readers by including data for a large number of cells and displaying the results in tSNE plots.

The tSNE plot is a universal tool used in almost every field of science and engineering. It is the clearest way to visualize high-dimensional data sets, like the multiplexed PLISH data set. We use k-means clustering to empirically define cell classes and their centroid expression profiles, but the tSNE plots are a useful complement because they efficiently display the number of cells in each class and the cell-to-cell variation within classes.

We collected data on a large number of cells because this capability is an important feature of PLISH. Unbiased identification of rare cell types requires a smISH approach with high cell-throughput.

The 'snapshot visualization' feature of PLISH is a downstream application enabling anatomical re-mapping of single cells or cell classes back onto the profiled tissue, and is independent of the tSNE plot.

8) Higher mag images are needed for Figure 3 to support the claim that this is "on par with IF" – the current images are not on par with IF and hardly support the presence of a putative BASC, while the IF in Figure 4—figure supplement 2E is much clearer.

By "on par with fluorescence immunostaining" we meant that the PLISH signal is strong enough that it can be easily visualized, facilitating applications like co-staining for a few transcripts (as shown in Figure 2, for example). We have clarified the language in the manuscript to reflect this. We have also added a close-up image of several AT2 cells and macrophages (Figure 2) and of 2 BASCs at an airway terminus (Figure 2), previously shown at low magnification. These micrographs demonstrate that overlaying PLISH stains in conjunction with DAPI allow discrimination of co-labeling in individual cells on par with the discrimination afforded by immunostaining.

9) Figure 3DE – what is the "other"? How does is fit in with what we already know, since these are a set of well-characterized genes? Should rework the text/figure in 3D to better lead into the application of this aspect of PLISH in Figure 4.

We have clarified the language in the manuscript on this point. We state that the "other" class consists of cells for which we did not include markers in our experiment. We also highlight that the four subsets within the general "other" class are defined by the differential expression of housekeeping genes, including one sub-class that is enriched in pulmonary arteries.

General comments:Overall, this is a highly technical paper that defines a tool. The authors clearly realize this, as they have filed a patent on PLISH. The concept is interesting and its application has great potential, yet it is not as excitingly novel as the text implies. While indeed an improvement on current technologies, this is fit for a methods report rather than a scientific article.

We agree that our manuscript could be viewed as a technical paper, and we have restructured it for consideration as a Tools and Resources report.

PLISH is excitingly novel because it enables fast expression profiling (>5 genes) in tissues. This addresses several current challenges, for example the comprehensive mapping of single-cell expression programs in intact tissue. PLISH thus provides a powerful complement to existing tissue-dissociative sc-RNAseq technologies.

We feel that *eLife* is an ideal open-access venue to make PLISH available for widespread use by the scientific community.

The data are confirmatory rather than exploratory, and serve mainly as tool validation.The focus on BASCs does not help the authors' case, as these are a controversial cell type not found in humans, and the IF images were more convincing than the PLISH data.

Our manuscript is not meant to address controversies about the functional significance of BASC cells in mice, or their significance to human biology. The BASC cells were called out simply as a test of whether PLISH could identify and map a known, transcriptionally-distinct cell type with a specific localization in the lung. We have added blown-up micrographs (Figure 2) that highlight the extremely convincing nature of PLISH data, particularly as it applies to BASC cells.

If the authors tackled more profound and subtle questions with PLISH, and in human samples (such as identifying the human BASC), then this would be less a tools paper and more a scientific article.

Our manuscript could be considered as a technical report, and we have restructured it for the Tools & Resourcesformat rather than the Research Articleformat.

Finally, the overall tone describing this technology is overly ambitious and exceedingly strong. While this is fantastic work at method development and optimization (as well as creating accessibility), it falls short of being an original research article. The cover letter overstates the novelty.

We are excited because we do feel PLISH is an original and very significant technological advance with immediate applications, which will directly impact our understanding of disease at the single-cell level in tissues.

Reviewer #2:[…] However, no biological insights are presented. For example, Figure 2 shows that the technique basically works in formalin-fized paraffin embedded human samples using extremely highly expressed transcripts, but these results had previously been obtained by immunostaining so no biological insights are presented.

We acknowledge the consensus among the reviewers that the manuscript did not include enough new biology for a Research Article. We have restructured our manuscript as a Tools and Resourcesarticle.

Note: the cell shape changes that are described in the text for SpC+ cells are not visible in the image on the merged pdf.

We have now included close-up images of SpC+ cells in both IPF and non-IPF lungs, which highlight the abnormal, flattened and elongated morphology (Figure 2).

The analysis of adult mouse lungs – Figure 3 and Figure 4 is a bit disappointing. This is solid proof-of-principle that the PLISH technique works. However, these transcripts are all highly expressed – no real biological insights are gained at all here. To publish in eLife I would at least expect to see low level transcripts detected, or some new biology, in addition to the proof of principal work shown here.

We have added new data demonstrating PLISH detection of the low-abundance Foxj1 RNA (Figure 2, Foxj1 is measured to have an FPKM value of 10 in single ciliated cells). This is in addition to the low-abundance Axin2 RNA that was used to calibrate the sensitivity of PLISH (Figure 1—figure supplement 3, FPKM value of 2), as reported in the original submission.

No prior evidence springs to mind as to what the 2 types of club cells could be – prox and dist, but this change in B-actin could be purely structural to do with cell size/shape in different parts of the airways.

We agree that the difference in transcriptional states between the two classes of Club cells is probably functional, for example due to different structural requirements in different parts of the airway. However, this type of data and insight is exactly what in situ expression profiling uniquely provides, since it would not emerge from tissue-dissociative scRNAseq approaches.

"Proper" analysis of a tissue section using probes to transcripts that are thought to be expressed at low levels, in addition to highly expressed genes such as Sftpc, is required. Plus some biological insight.

We have added a new figure panel showing PLISH detection of the low-abundance Foxj1 RNA (Figure 2, FPKM value of 10 in single ciliated cells). The PLISH detection of Axin2 RNA (Figure 1—figure supplement 3, FPKM value of 2) is another example. We have also now validated a novel AT1 cell type marker, Akap5, using PLISH (Figure 4—figure supplement 1).

Reviewer #3:The present study by the Desai lab shows the development of a proximity ligation in situ hybridization technology and its use to examine various cell type markers in the mouse and human lung. The technique appears to work well and if employed correctly could enhance studies on human tissue samples where histology is possible but isolation of single cells is limited. One limitation of the current study is that it does not extend our understanding of lung cell heterogeneity as most if not all of the markers examined have already been extensively characterized in the lung.

Given the consensus of the reviewers, we have re-classified our manuscript as a Methods report rather than an original Research Article. To address this specific comment, we have added data validating Akap5 as a new AT1 cell type marker (Figure 4—figure supplement 1). This demonstrates an important application of PLISH.

1) Much of the histology shown is at far too low of a magnification to assess the patterns of expression. The authors should provide high mag pictures in all situations, especially for the alveolar region as it is difficult to assess where the positive cells are located. Use of 3D reconvolution (i.e. IMARIS) would be helpful.

We apologize for the poor image quality, and have now included high-magnification micrographs and used dashed lines to mark bronchial and alveolar structures. We have tried to illustrate two distinct ways to apply PLISH: (i) direct visual examination of marker RNA overlays in a small field of cells (Figure 2) versus (ii) automated, computational analysis of multiple markers in a large number of cells (Figure 3). The former does not require any computational expertise. The latter does not require visual analysis by a human being with a high-level of biological expertise – but as we show, accurately distinguishes cell types.

2) It would be nice for the authors to show data from some new novel markers that were isolated from the current study. This would provide novel insight into how this technique has advantages over other techniques.

We have added the validation of a new AT1 cell type marker inferred from a prior scRNAseq analysis (Akap5, Figure 4—figure supplement 1).

We note that the key advance of PLISH over existing techniques (RNAScope, ViewRNA, Stellaris, HCR etc.) is that it enables fast multiplexing in intact tissue. We have focused the manuscript on demonstrating this aspect of the technology.

3) A lot of the co-staining is difficult to assess. These data may be helped by higher mag imaging.

We apologize for the poor quality of the images in the original submission. We have substituted higher magnification images in the revised manuscript, and also incorporated dashed lines to demarcate lung structure for easier interpretation.

4) In the scRNA-seq experiment, was this done on whole lung cell suspension or Epcam+? It is not mentioned in the paper that I can see.

The scRNA-seq data was from a previously published manuscript that employed depletion of CD45+ cells and enrichment of Epcam+ cells from dissociated mouse lung with the use of magnetic bead-activated cell sorting (Treutlein et al., Nature, 2014).

5) Many of the figures need better annotation. It’s very hard to interpret what some of the panels are trying to convey.

We thank the reviewer for this feedback. We have improved figure annotation, added outlines of anatomical structures, provided higher resolution and close-up micrograph images, and revised the legends for clarity.